# Tunable Transparency and NIR-Shielding Properties of Nanocrystalline Sodium Tungsten Bronzes

**DOI:** 10.3390/nano11030731

**Published:** 2021-03-14

**Authors:** Luomeng Chao, Changwei Sun, Jianyong Dou, Jiaxin Li, Jia Liu, Yonghong Ma, Lihua Xiao

**Affiliations:** 1College of Science, Inner Mongolia University of Science and Technology, Baotou 014010, China; 2017993@imust.edu.cn (L.C.); 18722419110@163.com (C.S.); djyylwl880724@163.com (J.D.); ljx19970319@163.com (J.L.); jialiu@imust.edu.cn (J.L.); 2Guizhou Institute of Technology, Guiyang 550003, China

**Keywords:** sodium tungsten bronzes, nanoparticles, tunable optical properties

## Abstract

The Na*_x_*WO_3_ nanoparticles with different *x* were synthesized by a solvothermal method and the absorption behavior in visible and near-infrared light (NIR) region was studied. Well-crystallized nanoparticles with sizes of several tens of nanometers were confirmed by XRD, SEM and TEM methods. The absorption valley in visible region shifted from 555 nm to 514 nm and the corresponding absorption peak in NIR region shifted from 1733 nm to 1498 nm with the increasing x. In addition, the extinction behavior of Na*_x_*WO_3_ nanoparticles with higher *x* values were simulated by discrete dipole approximation method and results showed that the changing behavior of optical properties was in good agreement with the experimental results. The experimental and theoretical data indicate that the transparency and NIR-shielding properties of Na*_x_*WO_3_ nanoparticles in the visible and NIR region can be continuously adjusted by *x* value in the whole range of 0 < *x* < 1. These tunable optical properties of nanocrystalline Na*_x_*WO_3_ will expand its application in the fields of transparent heat-shielding materials or optical filters.

## 1. Introduction

Tungsten bronze is a kind of solid solution formed by other cations filling in the lattice structure of WO_3_. Its chemical formula can be written as M*_x_*WO_3_, in which M is dopant cation and the *x* value can vary in a certain range (0 < *x* < 1, M is typically electropositive metals such as alkali, alkaline earth or rare-earth metals), which are non-stoichiometric compounds. The dopant M cation can contribute a number of electrons to reduce part of the hexavalent tungsten to pentavalent, which makes tungsten bronzes have special physical and chemical properties such as superconductivity [1], photochromism [2], electrochromism [3], photothermal conversion [4] and transparent heat-shielding properties [5], etc. Among these properties, the transparent heat-shielding properties have been studied extensively in recent years because the tungsten bronze exhibits low absorption of visible light and high absorption of near-infrared light (NIR), which meets the demand of smart windows with high visible transmittance and heat-shielding performance. The free electrons in tungsten bronzes can be regarded as the electron gas moving under the background of uniform positive charge (actually a kind of plasma). The plasma may resonate when it meets the incident light, while the plasma resonance frequency increases with the increase of carrier concentration and moves to the short-wave direction.

Tungsten bronzes have three types of cubic, tetragonal and hexagonal phases [5], and the tunnels formed by the WO_6_ octahedra with different phases are also different. The cubic phase contains only one type of cubic cavity, while the tetragonal phase contains not only tetragonal, but also tripartite and pentagonal channels. The hexagonal structure, which has been widely studied, contains tripartite and hexagonal channels. The size of dopant cation M determines the content and position of M in the structure of tungsten bronze. H^+^ and smaller alkali metal ions (such as Li^+^) can locate in the narrow tripartite channel, while larger alkali metal ions (such as K^+^, Cs^+^) or NH^4+^ can only locate in the hexagonal channel. If the dopant cation occupies all the hexagonal channels, then *x* = 0.33. Among the various M*_x_*WO_3_, many studies indicate that Na*_x_*WO_3_ is the only compound which is possible to be synthesized in the wide range of 0 < *x* < 1 [6,7,8], so it has become the most studied tungsten bronze. The electrical and optical properties of Na*_x_*WO_3_ can substantially change with varying Na content. When *x* < ∼0.2, Na*_x_*WO_3_ exhibits semiconductor properties, and the electrical conductivity increases with increasing *x* and thus Na*_x_*WO_3_ shows metallic properties at high *x* [9]. Moreover, the changing electrical conductivity also leads to changing color in Na*_x_*WO_3_. As the *x* increases from 0 to 1, the color of Na*_x_*WO_3_ gradually changes from lime green to dark blue, violet, pink, orange and yellow [10]. This change in color can be attributed to increase of bulk plasma frequency (*ω*_p_) [6]. Unlike other metals, the *ω*_p_ of Na*_x_*WO_3_ can be tuned by Na content in WO_3_ [11,12].

In our previous work, we found that extinction behavior in visible and NIR region of rare-earth hexaboride (RB_6_) is directly correlated with its *ω*_p_, and nanocrystalline RB_6_ shows tunable optical characteristic as an excellent transparent heat-shielding material [13]. Like RB_6_, nanocrystalline M*_x_*WO_3_ is also an excellent transparent heat-shielding material, and even has a wider absorption range in the NIR region than LaB_6_ [14]. Because of the tunable *ω*_p_ characteristic of Na*_x_*WO_3_, we infer that the extinction in the visible and NIR region of nanocrystalline Na*_x_*WO_3_ is also tunable by different Na content. However, the current research on tungsten bronzes mainly focuses on its preparation methods or co-doping effect [5,15,16,17,18]; the tunable absorption behavior of nanocrystalline tungsten bronzes has very rarely been reported in the literature. In this paper, nanocrystalline Na*_x_*WO_3_ powders with different *x* were synthesized by a solvothermal method and their optical properties were discussed. In order to more systematically study the optical properties of nanocrystalline Na*_x_*WO_3_, the discrete dipole approximation (DDA) method was also used to investigate the influence of different x, particle size and particle shape on its optical properties.

## 2. Materials and Methods

### 2.1. Fabrication

To fabricate Na*_x_*WO_3_ nanopowders, different amounts (0.09 g, 0.27 g, 0.45 g and 0.63 g, respectively) of NaOH particles (Tianjin Kemiou Chemical Reagent Co., Tianjin, China) were fully ground and added gradually into 150 mL benzyl alcohol, then mixed with WCl_6_ (Shanghai Macklin Biochemical Co., Ltd., Shanghai, China) by magnetic stirring to obtain a dark blue precursor solution. In the precursor solution, the concentration of WCl_6_ remained at 0.015 M. Subsequently, the precursor solution was transferred into Teflon-lined autoclave, and reacted at 200 °C for 4 h. After natural cooling, the reactants were washed with pure water and ethanol several times, then centrifuged and dried in vacuum at 40 °C for 1 h to obtain blue powder of Na*_x_*WO_3_.

### 2.2. Characterization

The phase identification of the samples was analyzed by X-ray diffraction (XRD, Philips PW1830, Hohhot, Inner Mongolia, China) with Cu Kα (λ = 1.5406 Å), at 30 kV voltage with 30 mA current and scanning rate of 1°/min. The sample morphology was examined by using a field emission scanning electron microscope (SEM, Gemini 300, Beijing, China) with an Energy Dispersive Spectrometer (EDS, Oxford X-Max, Beijing, China). The microstructure was further characterized by transmission electron microscopy (TEM, FEI Talos F200X, Beijing, China) with accelerating voltage of 200 kV. The valence state of elements was analyzed by X-ray photoelectron spectroscopy (XPS, Escalab 250Xi, Baotou, Inner Mongolia, China) using a monochromatic Al kα X-ray source. Optical measurements of samples were performed at room temperature by using ultraviolet–visible–near infrared spectrometer (UV-Vis, HITACHI UH4150, Baotou, Inner Mongolia, China).

### 2.3. Simulation Method

Discrete dipole approximation method (DDA) was used to study the influence of different x, different size and different shape on the extinction behavior of Na*_x_*WO_3_ nanoparticles. DDA is a method to calculate the far-field and near-field optical properties of nanostructures (usually called “target”) with a variety of shapes and sizes, which adopts a finite array of dipoles to approximate the continuous material. The dipoles polarize under the action of light field and interact with each other through electric field. The absorption and scattering of materials can be obtained by calculating the polarization of the dipoles. Our previous work has proved that DDA method has strong advantages in calculation of transparent heat-shielding materials [19].

In the present work, DDSCAT7.3 software was used to simulate all the DDA results. DDSCAT7.3 is an open source fortran-90 program developed by Drain and Flatau of Princeton University to calculate the scattering and absorption of electromagnetic waves by particles with various geometric shapes and complex refractive indexes [20,21]. The extinction efficiency of target can be expressed as
$$Q_{ext}=\frac{C_{ext}}{\pi a_{eff}^{2}}$$
where *C_ext_* is the extinction cross section and *a_eff_* is the effective radius (a radius of an equal volume sphere) of target. The complex dielectric constant of Na*_x_*WO_3_ measured by Owen et al. was used to calculate the extinction efficiency in this work [22].

## 3. Results and Discussion

To obtain a nanocrystalline Na*_x_*WO_3_ with different *x* value, we mixed precursor solutions with different molar ratio of Na/W, as shown in Figure 1. After magnetic stirring at 30 min, the solution with Na/W molar ratio of 1:1, 3:1 and 5:1 turned dark blue, but the solution with molar ratio of 7:1 was always kept at yellowish. After treatment in the reactor, blue powder was obtained from the solution with Na/W molar ratio of 1:1, 3:1 and 5:1, but no solid precipitate was formed in the solution with Na/W molar ratio of 7:1. We infer that tungsten trioxide reacted with NaOH and formed sodium tungstate when NaOH was excessive, so other methods are needed to obtain nanosized Na*_x_*WO_3_ with high *x* value.

Figure 2 shows the XRD results of three kinds of Na*_x_*WO_3_ nanopowders obtained from the solvothermal reaction. The sharp and intensive XRD reflections indicate the well-crystallized character of the samples. The XRD pattern of sample with Na/W molar ratio of 1:1 can be indexed on the basis of a hexagonal phase of Na-WO_3_ (JCPDS 81-0577). With increasing Na content, the peaks of (001) and (002) become stronger and all other peaks become weaker. The peaks of (001) and (002) also correspond to a cubic phase of Na-WO_3_ (JCPDS 28-1156). In addition, it can be seen from the partial enlarged view that the (001) peak slightly moves to higher angle with the increase of Na content, which corresponds to the position of the peak near 23° on the JCPDS 81-0577 and JCPDS 28-1156. Therefore, we infer that the sample exhibits a tendency to change to cubic phase with increasing Na content. According to literature, the Na*_x_*WO_3_ shows a perovskite-type crystal structure with cubic symmetry at *x* > ~0.4 [6,7], which is consistent with our XRD results.

The SEM images of the synthesized Na*_x_*WO_3_ powders with different Na/W ratios are shown in Figure 3. It can be seen from Figure 3a–c that all three samples are mainly composed of homogeneous and well-dispersed nanoparticles with sizes of several tens of nanometers. The EDS images in Figure 3d–f gives that the atomic ratios of Na/W for three samples are 0.077, 0.173 and 0.243, respectively. Figure 3g–i show the element mapping of sample with Na/W ratio of 1:1, which confirms the presence and uniform distribution of O, W and Na in selected area.

To further study the detailed microstructure of obtained products, TEM was used to observe the grain morphology and crystallinity of the sample with Na/W molar ratio of 1:1, and results are given in Figure 4. It can be seen from the TEM image that the sample is composed of nanoparticles with sizes of several tens of nanometers, which is consistent with the results of SEM. Moreover, two kinds of crystalline lattice constant in the HRTEM images were calculated as 0.389 and 0.638 nm, which agrees well with the interplanar spacing (0 0 1) and (100) of the hexagonal phase of Na-WO_3_ (JCPDS 81–0577), respectively.

The chemical state of the Na*_x_*WO_3_ nanoparticles was carefully determined by XPS. Figure 5 shows the typical XPS of the tungsten core level (W_4f_) in the Na*_x_*WO_3_ nanoparticles with Na/W ratio of 1:1. The spectrum can be fitted to two groups of spin-orbit doublets of W_4f7/2_ and W_4f5/2_ with a separation distance of 2.1 eV, indicating two different oxidation states of W element. The peaks at 37.6 and 35.5 eV can be attributed to the W element being in a 6+ oxidation state, while 36.5 and 34.4 eV can be assigned to the W element being in a 5+ oxidation state. It is suggested that the typical nature of non-stoichiometric tungsten bronzes can be expressed as the formula of M*_x_*W^6+^_1-*x*_W^5+^*_x_*WO_3_; our XPS results are in good agreement with this reduced feature. Na atoms can contribute a number of free electrons when they are doped into the structure of WO_3_, and part of the W^6+^ will be reduced to W^5+^. The transparent heat-shielding properties of tungsten bronzes are closely related to the plasmon resonance of free electrons, and the concentration of free electrons has a great influence on the *ω*_p_. Therefore, the transparent heat-shielding properties of tungsten bronzes could be tuned by controlling the concentration of free electrons in its microstructure.

To investigate the optical properties of Na*_x_*WO_3_, the same amount of obtained powder sample was uniformly dispersed and coated on the glass slide, and the absorption spectra are shown in Figure 6. All three samples exhibit low and high absorption characteristics in the visible and NIR region, respectively. The absorption valley occurs at 555 nm for the sample with Na/W ratio of 1:1, and shifts to shorter wavelength of 514 nm for the sample with Na/W ratio of 5:1. The corresponding absorption peak in NIR region shows the same trend as the absorption valley, with shifts from 1733 nm to 1498 nm. These results illustrate that the transparency in visible and NIR region of nanocrystalline Na*_x_*WO_3_ can be effectively tuned by *x* content. Generally, the low optical absorption in visible region can be ascribed to the *ω*_p_, which corresponds to the peak in low energy region of energy loss spectra. The electron energy loss spectra for Na*_x_*WO_3_ with different *x* value have been measured by Tegg et al. [12], and results showed that the position of peak in low energy region shifts to higher energy as *x* increases, which is consistent with our results in Figure 6.

However, the nanoparticulated Na*_x_*WO_3_ with higher *x* values could not be synthesized in our experiment. Therefore, the optical properties of nanosized Na*_x_*WO_3_ with higher *x* values in the visible and NIR region were calculated theoretically by using the DDA method. The DDA simulation gives the extinction efficiencies (*Q_ext_*) of single particle. In order to compare simulation results with the experiment, we plotted *Q_ext_*/*a_eff_* [23]. It should be noted that the absorption curve measured by UV-Vis spectrophotometer is converted from the transmittance of the sample, so the absorption spectrum from our measurement corresponds to the extinction curve from DDA calculations. Figure 7 gives the extinction curves of sphere-shaped Na*_x_*WO_3_ particles with size of 50 nm and *x* value of 0.522, 0.628, 0.695, 0.740, 0.860 and 0.940. For *x* = 0.522, the extinction valley occurs at 501 nm and gradually shifts to 423 nm for *x* = 0.94 with increasing x, which is in good agreement with our experimental results. Combining the experimental results with the DDA simulation results, it can be found that the position of the transmission peak of Na*_x_*WO_3_ in the visible region can be continuously adjusted by *x* value in the whole range of 0 < *x* < 1. In addition, the extinction peak position shifts from 749 nm to 640 nm when *x* increases from 0.522 to 0.94. Although the shifting trend with *x* value is the same as that in Figure 6, the position of extinction peak is much smaller than that in Figure 6. The reason for this difference should be related to the size and shape of nanoparticles. The experimental sample is composed of particles of various sizes and shapes, and the measured absorption spectrum is the comprehensive effect of all particles. However, what DDA calculates is particles with size of 50 nm and shape of ideal sphere. In order to verify this inference, we also calculated the extinction behavior of Na*_x_*WO_3_ particles of different sizes and shapes.

The calculated extinction curves of spherical-shaped Na_0.522_WO_3_ particles with different sizes and differently shaped Na_0.522_WO_3_ particles with size of 50 nm are given in Figure 8. It can be found that there was significant difference between the extinction curves of different-sized particles. The extinction valley shifts toward short wavelength direction and broadens with the decreasing particle size. While the extinction peak broadens and shifts to long wavelength direction with the increasing particle size, the peak intensity weakens. For different shapes, there is little difference in the shape and position of the extinction valley, while the intensity, width and position of extinction peak are very different. Generally, a strong absorption or scattering effect occurs when the collective oscillation frequency of the conduction electrons is the same as that of the incident photons, which leads to an enhancement of the electromagnetic field in a very small area of the particle surface. On the other hand, the charge distribution produced by collective oscillation of particles with different shapes is also different; our calculated electric field distribution around the variously shaped Na*_x_*WO_3_ particles is shown in Figure 9 (all simulations assume that the incident light propagates along the x-axis.). This suggests that the localized surface plasmon resonance (LSPR) is highly sensitive to different symmetries and angles. Therefore, the extinction behavior of particles with different shapes is quite different in the near infrared region. From the simulation results of particles with different sizes and shapes, it can be concluded that the wide range absorption in the NIR region in the experiment is a common effect of various sizes and shapes of Na*_x_*WO_3_ particles.

## 4. Conclusions

The optical properties of nanoparticulated Na*_x_*WO_3_ with different *x* have been both experimentally and theoretically investigated in this paper. The EDS results showed that the atomic ratios of Na/W for three samples are 0.077, 0.173 and 0.243, indicating that *x* is gradually increasing in the synthesized samples. Absorption measurement showed that the absorption valley in visible region shifts from 555 nm to 514 nm when Na/W ratio changes from 1:1 to 5:1, and the corresponding absorption peak in NIR region shifts from 1733 nm to 1498 nm. These show that the absorption (or transmittance) behavior in visible and NIR region of nanocrystalline Na*_x_*WO_3_ can be effectively tuned by *x* content. The subsequent DDA simulation results show that this tunable characteristic is also suitable for the nanoparticulated Na*_x_*WO_3_ with higher *x* values. The continuously tunable optical properties of nanocrystalline Na*_x_*WO_3_ make it very promising in the fields of transparent heat-shielding materials or optical filters.

## Figures and Tables

**Figure 1 nanomaterials-11-00731-f001:**
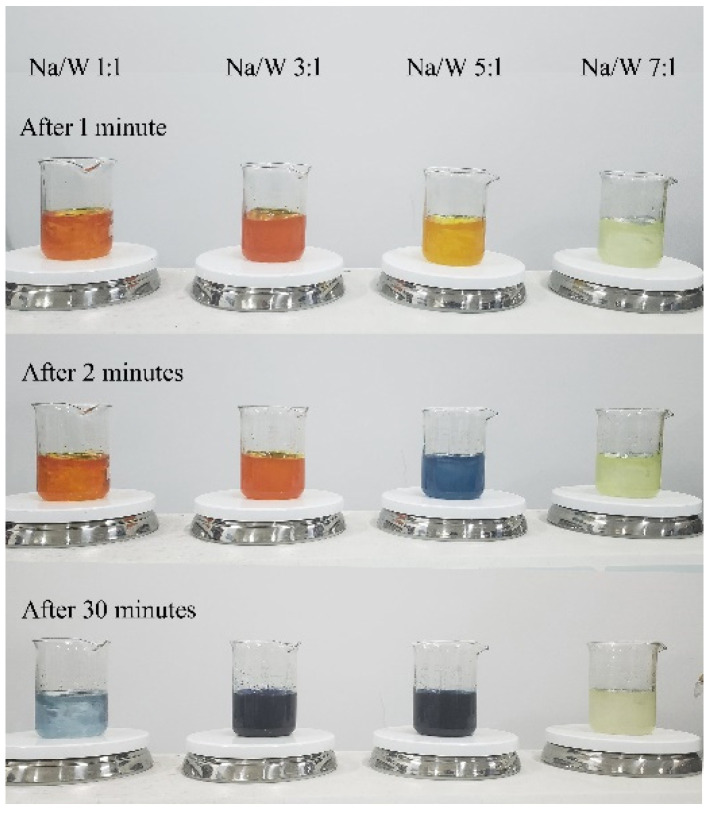
As-prepared precursor solutions with different molar ratio of Na/W after different times.

**Figure 2 nanomaterials-11-00731-f002:**
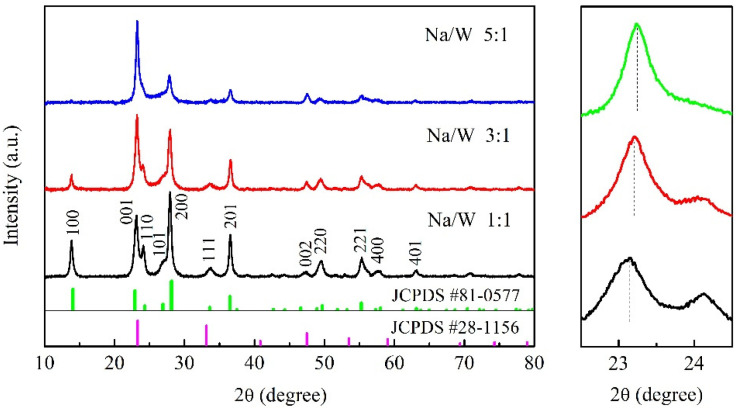
XRD patterns of the obtained powders (**left**) and partial enlarged view (**right**).

**Figure 3 nanomaterials-11-00731-f003:**
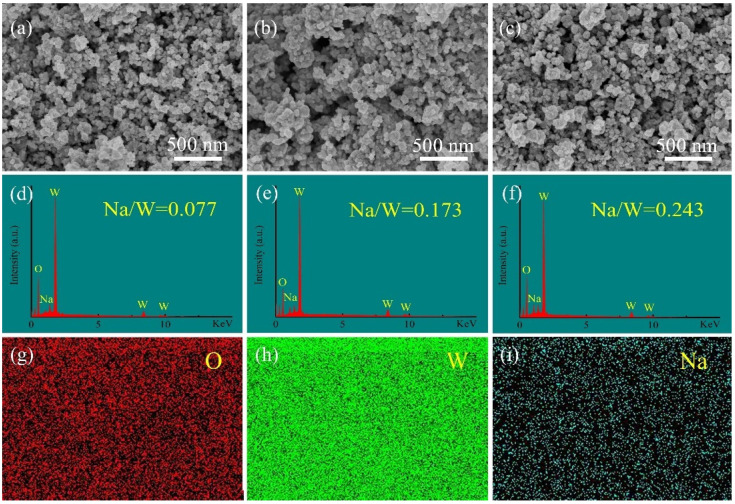
SEM images of Na*_x_*WO_3_ synthesized with the Na/W ratio of (**a**) 1:1, (**b**) 3:1, (**c**) 5:1; corresponding EDS images (**d**) 1:1, (**e**) 3:1, (**f**) 5:1; element mapping of sample with Na/W ratio of 1:1 (**g**) O, (**h**) W, (**i**) Na.

**Figure 4 nanomaterials-11-00731-f004:**
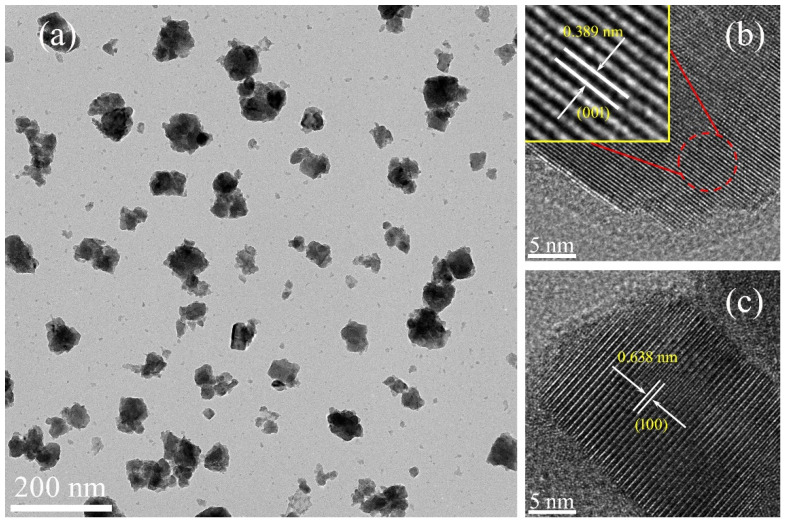
TEM image (**a**) and HRTEM images (**b**,**c**) of the Na*_x_*WO_3_ nanoparticles with Na/W ratio of 1:1.

**Figure 5 nanomaterials-11-00731-f005:**
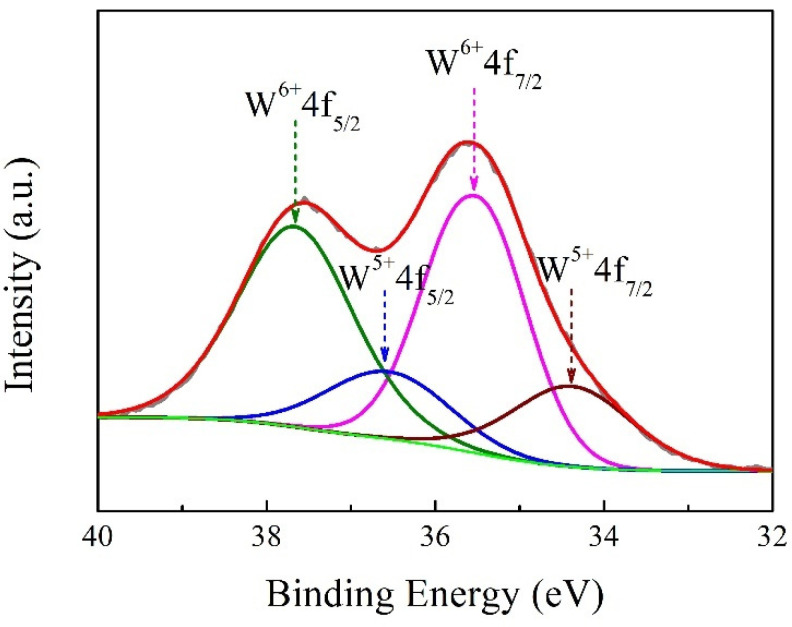
Tungsten core-level (W_4f_) XPS spectra of the Na*_x_*WO_3_ nanoparticles with Na/W ratio of 1:1.

**Figure 6 nanomaterials-11-00731-f006:**
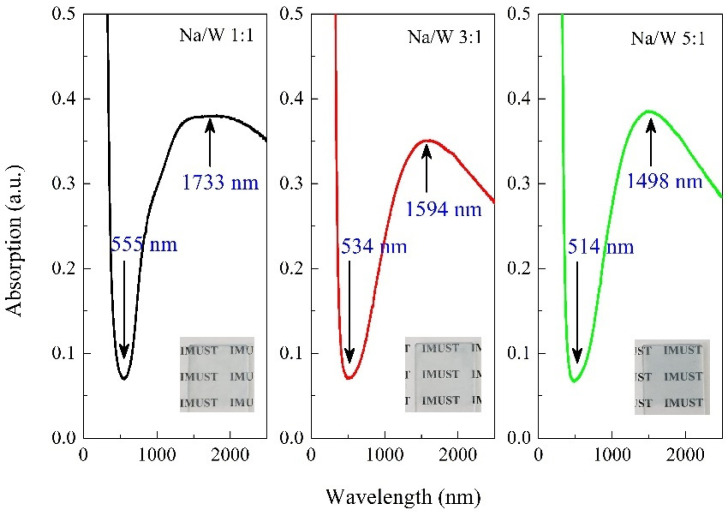
Absorption spectra of Na*_x_*WO_3_ nanoparticles with Na/W ratio of 1:1, 3:1 and 5:1. Inset shows the photograph of the thin film samples used in the test.

**Figure 7 nanomaterials-11-00731-f007:**
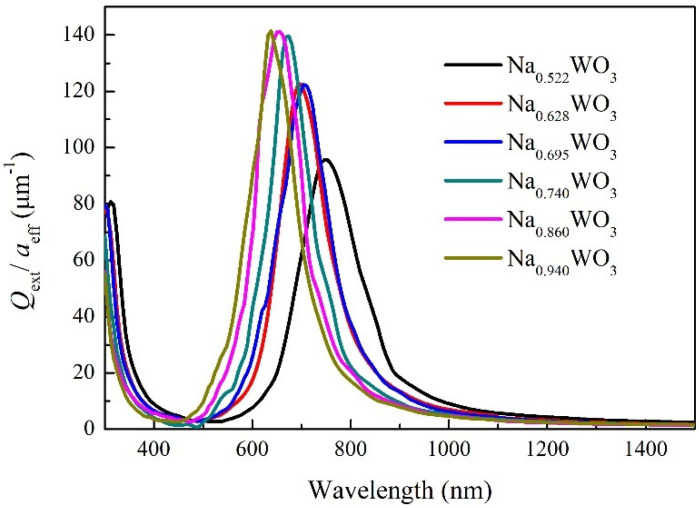
Extinction curves of sphere-shaped Na*_x_*WO_3_ particles with size of 50 nm and different *x* value.

**Figure 8 nanomaterials-11-00731-f008:**
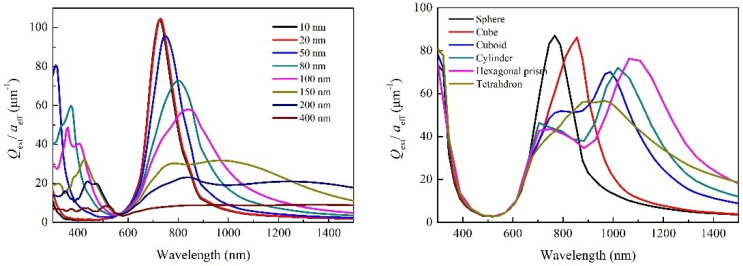
Extinction curves of spherical-shaped Na_0.522_WO_3_ particles with different sizes (**left**) and differently shaped Na_0.522_WO_3_ particles with size of 50 nm (**right**).

**Figure 9 nanomaterials-11-00731-f009:**
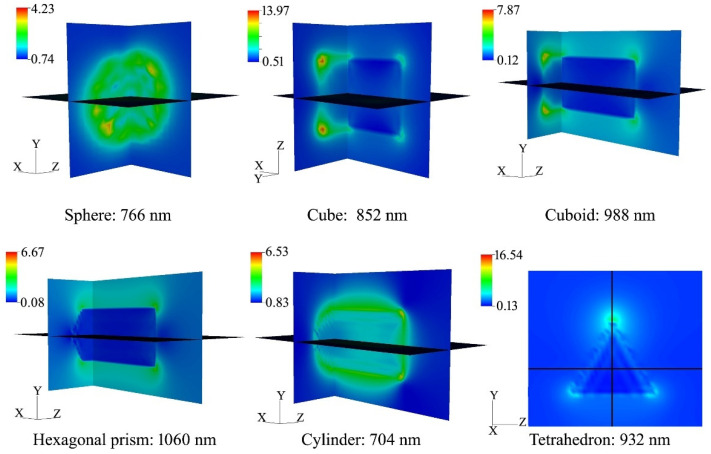
Electric field distribution around the variously shaped Na_0.522_WO_3_ particles with effective radius of 50 nm at their corresponding plasma resonance peaks.

## Data Availability

The data presented in this study are available on request from the corresponding author.

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
