# Peer review of "Tunable Transparency and NIR-Shielding Properties of Nanocrystalline Sodium Tungsten Bronzes"

_nanomaterials, 2021, doi:10.3390/nano11030731_

Round 1
Reviewer 1 Report
Although the article is of some particular interest, in this form it is not suitable for publication and needs major revision.
- The introduction is not written clearly enough, the position of this study among similar ones is not at all clear. Moreover, the authors are silent or unfamiliar with similar recent studies. See for example, some recent papers and references therein on NIR shielding:
- a) Ceramics International, Volume 47, Issue 6, 15 March 2021, Pages 8345-8356. Tm, Yb co-doped urchin-like CsxWO3 nanoclusters with dual functional properties: Transparent heat insulation performance and enhanced photocatalysis(Article)
Yang, J., et al https://doi.org/10.1016/j.ceramint.2020.11.197
- b) Aerosol and Air Quality Volume 20, Issue 4, April 2020, Pages 690-701
Aerosol-assisted production of NIR shielding nanoparticles: Sodium tungsten bronze(Article)
Tu, H., Wang, W., Chen, D.-R. https://doi.org/10.4209/aaqr.2019.10.0548
- c) Applied Physics A: Materials Science and ProcessingVolume 126, Issue 2, 1 February 2020, 98
Synthesis of cesium tungsten bronze by a solution-based chemical route and the NIR shielding properties of cesium tungsten bronze thin films(Article)
Wu, P.-J.a, et al https://link.springer.com/article/10.1007/s00339-020-3291-4
- Ceramics International, Volume 44, Issue 12, 15 August 2018, Pages 13469-13475 Synthesis of CsxWO3 nanoparticles and their NIR shielding properties(Article)
Yao, Y., et al https://doi.org/10.1016/j.matlet.2020.127847
- Excessive self-citation of their own papers on borides to the detriment of articles on MxWO3 is not a good fact and should be corrected.
- It would be useful to make a comparison with other materials, as done in [5], adding new reference sources,
- It would be useful to report that the authors managed to find out about the degradation/long term stability of reported nanocrystalline NaxWO3
- Figure 9 is not quite clear and could be improved. while in Fig.2 we can see “patterns” as well as spectra in Fig. 5, 6.
Reviewer 2 Report
Chao et al report a study of sodium-tungsten bronzes in nanoparticle form. They discuss structure, morphology and optical response, and find an explanation for the IR absorption and VIS transparency of their system. Their manuscript is interesting and reports findings of relevant fundamental and applicative interest. Before it can be recommended for publication, however, some amendments have to be made.
- after line 52-53, the authors should stop using the expression MxWO3 and systematically replace "M" with "Na"
- is it possible to extract a size/shape distribution from the TEM images of Fig.4, to feed in the DDA calculations? In the Image, the distribution of particle size seems to include systems of very different size. Are the largest particles single-crystal structures? Or are they agglomerates of different crystals? Does the distribution depend on how the powders are dispersed?
- for the DDA simulations, a dielectric function for the bronze must be inserted in the calculations. Where did the authors obtain it? Are they sure these values, if obtained from the literature, are compatible with their samples?
Reviewer 3 Report
The article of Chao et al concerns the synthesis by a solvothermal technique, the characterization and the spectroscopic study of sodium tungsten bronzes, "old materials" but with new potential applications as heat shielding materials or optical fibers.
In my opinion this manuscript deserves to be published as it shows original and interesting results. However to improve ther quality of the paper I think corrections are needed concerning the following items:
- Introduction line 45: I think the tungsten bronze cubic polymorph shows cubic cavities, not square channels, since interstitial sites are surrounded by [WO6] octahedra.
- Line 50: strictly speaking NH4+ is not an alkali cation.
- line 52: I think the correct sentence is "Among the various MxWO3, many studies indicate that NaxWO3...". By considering this, ref 8 does not concern Na materials.
- Line 53: according to refs 6 and 7, the maximum x value is respectively 0.8 and 0.93, so I do not think that the "full range" is obtained.
- Lines 74-77: this sentence already concerns results and must be moved.
- Section 2.1: the synthesis protocol must specify the different tested Na/W ratios (given later in lines 121-126). Are the blue solution and powder a sign that the reaction worked?
- Fig.2: the XRD pattern should start at 10° for the Na/W 1:1 hexagonal form since a (110) peak exists before 20°.
- According to literature concerning the cubic form, the XRD patterns are characterized by two main peaks located near 23 and 33°. However the 33° peak is not observed in Na/W 5:1 pattern. Therefore how can authors conclude that the cubic form is obtained (line 140)? Please give the JCPDS 28-1156 in Fig. 2.
- According to Fig.3 the Na/W ratio does not reach the 0.4 value (line 142) for which a cubic form is observed. There is an inconsistency between EDS and XRD results concerning the cubic form.
- Fig. 5: the color codes are wrong. Give the 4f7/2-4f5/2 assignment for each W6+ and W5+ cation.
- Fig. 6: is the samples colour related to these absorption spectra? please explain.
- Line 211: How are these x values chosen?
- How is the Cext (extinction cross section) value found?
- Conclusion line 263: it is better to report the Na/W ratio in final samples, not in the synthesis batch.
- ref 24: the correct year is 2010.
Round 2
Reviewer 1 Report
Revised version is OK, so now this manuscript can be accepted
Reviewer 2 Report
The manuscript in its present form can be accepted for publication.
Reviewer 3 Report
The article of Chao et al can be now accepted for publication